# Effect of Electrochemically Active Top Electrode Materials on Nanoionic Conductive Bridge Y_2_O_3_ Random-Access Memory

**DOI:** 10.3390/nano14060532

**Published:** 2024-03-16

**Authors:** Yoonjin Cho, Sangwoo Lee, Seongwon Heo, Jin-Hyuk Bae, In-Man Kang, Kwangeun Kim, Won-Yong Lee, Jaewon Jang

**Affiliations:** 1School of Electronic and Electrical Engineering, Kyungpook National University, Daegu 41566, Republic of Koreasangw98@knu.ac.kr (S.L.); pos03034@knu.ac.kr (S.H.); jhbae@ee.knu.ac.kr (J.-H.B.); imkang@ee.knu.ac.kr (I.-M.K.); 2School of Electronics Engineering, Kyungpook National University, Daegu 41566, Republic of Korea; 3School of Electronics and Information Engineering, Korea Aerospace University, Goyang 10540, Republic of Korea; kke@kau.ac.kr; 4The Institute of Electronic Technology, Kyungpook National University, Daegu 41566, Republic of Korea

**Keywords:** sol–gel, Y_2_O_3_, RRAM, CBRAM, top electrodes

## Abstract

Herein, sol–gel-processed Y_2_O_3_ resistive random-access memory (RRAM) devices were fabricated. The top electrodes (TEs), such as Ag or Cu, affect the electrical characteristics of the Y_2_O_3_ RRAM devices. The oxidation process, mobile ion migration speed, and reduction process all impact the conductive filament formation of the indium–tin–oxide (ITO)/Y_2_O_3_/Ag and ITO/Y_2_O_3_/Cu RRAM devices. Between Ag and Cu, Cu can easily be oxidized due to its standard redox potential values. However, the conductive filament is easily formed using Ag TEs. After triggering the oxidation process, the formed Ag mobile metal ions can migrate faster inside Y_2_O_3_ active channel materials when compared to the formed Cu mobile metal ions. The fast migration inside the Y_2_O_3_ active channel materials successfully reduces the SET voltage and improves the number of programming–erasing cycles, i.e., endurance, which is one of the nonvolatile memory parameters. These results elucidate the importance of the electrochemical properties of TEs, providing a deeper understanding of how these factors influence the resistive switching characteristics of metal oxide-based atomic switches and conductive-metal-bridge-filament-based cells.

## 1. Introduction

Until now, several efforts have been made to push flash memory technology to its physical downscaling limit. Extremely downscaled devices show low power consumption and high performance, and can also enable the realization of a highly integrated system for future applications. However, they can face limitations [1]. To resolve these matters, resistive random-access memory (RRAM) devices have been considered a promising candidate to replace conventional flash memory as a next-generation memory model [2,3,4,5]. This is because they have an uncomplicated metal–channel–metal structure, a destruction-free readout, fast operation, low energy consumption, and high endurance. Many promising metal oxides have been explored as active main channel materials between two electrodes, such as SiO_x_, ZrO_2_, TiO_x_, Hf_x_O, and Y_2_O_3_ [6,7,8,9,10,11]. Among them, Y_2_O_3_ plays an important role due to its promising properties such as its large dielectric constant (k = 14–18) and optical bandgap (E_g_ = ~5.5 eV). These properties make Y_2_O_3_ useful as high-k insulators in complementary metal–oxide–semiconductor (CMOS) processes with low-k SiO_2_ insulators [12,13]. In addition, good chemical, mechanical, and thermal stability renders Y_2_O_3_ a potential candidate for surface coating layers in harsh environments [14]. Its high refractive index (n = 2.1) can make it a waveguide in solid state lasers [15,16]. Particularly for RRAM devices, Y_2_O_3_ has high defect chains along preferred orientations, which is helpful for easily forming the conductive filament, and leads to forming process-free RRAM devices [17,18]. There are two types of RRAM corresponding to conductive filaments between two electrodes: oxygen vacancy-based conductive filament-based OxRRAM and conductive-metal-bridge-filament-based conductive bridging random access memory (CBRAM) [19,20,21]. In the case of CBRAM, to form the conductive metal filament, active metal top electrodes (TEs), such as Cu, are used. This is more advantageous for integrated circuits in the CMOS industry because it suppresses the electron migration or resistor-capacitor (RC) delay compared to OxRRAM [22,23]. In the case of CBRAM, electrodes exhibiting electrochemical activity, such as Ag or Cu, and those that are electrochemically inert, such as Au or Pt, are utilized. The conductive filaments are formed via electrochemical reactions and diffuse inside the active RRAM channel materials. Finally, the reformation process occurs at the electrochemically inert bottom electrodes (BEs) from diffused metal ions. From that moment, conductive metal filaments grow from the inert electrodes. By making a short circuit between two electrodes, the resistance state changes from the high-resistance state (HRS) to the low-resistance state (LRS). The formed conductive metal filaments are broken via applying a negative voltage to the reduction process, which is slightly assisted by Joule heating. Finally, the HRS is attained. Electrochemically active electrodes have their own physical and electrochemical properties, such as oxidation rate, the number of generated mobile metal ions, migration speed, and the redox rate at the electrochemically inert BEs. Therefore, different electrochemically active electrodes can affect RRAM characteristics, such as the SET and RESET voltages, HRS and LRS values, endurance, and retention.

In this research, Y_2_O_3_-based RRAM devices were fabricated. The active channel layer, Y_2_O_3_, was formed on ITO electrodes using the sol–gel process. For the TEs, Au, Ag, and Cu were used. The electrical properties of the Y_2_O_3_-based RRAM devices prepared via the sol–gel process with different representative electrodes were investigated. The fabricated devices with electrochemically inert Au TEs do not exhibit conventional RRAM characteristics. In contrast, the fabricated devices with electrochemically active Ag or Cu show conventional bipolar RRAM characteristics, and there is no requirement for a high-voltage forming process. Between Ag and Cu, Ag ions from Ag electrodes migrate faster inside Y_2_O_3_, which leads to a decrease in the SET voltage, as well as easy conductive filament formation and thick conductive filament formation. The thicker conductive filament also makes it difficult to break the formed conductive filament, resulting in larger negative RESET voltages. However, the easy formation of conductive filaments successfully increases the number of programming–erasing cycles, i.e., endurance, when compared to electrochemically active Cu TEs.

## 2. Materials and Methods

To study the impact of different TEs on the properties of Y_2_O_3_ RRAM, devices were fabricated using three distinct TEs: Ag, Cu, and Au. The structure of the devices used in this experiment was glass/ITO/Y_2_O_3_/TE. Commercial deposited glass substrates were used (surface resistivity: 70–100 Ω/sq; slide; Sigma-Aldrich, St. Louis, MO, USA) ITO-coated glass substrates were cleaned for 10 min each, first with acetone and then with deionized water. The ultraviolet/O_3_ step (UV lamp wavelength: 254 nm; photon flux density: 16 mW/cm^2^; SEN Lights SSP16-110, Osaka, Japan) was performed for 1 h to enhance the hydrophilicity of the substrate surface and remove unnecessary organic residues. In 5 mL of 2-methoxyethanol (anhydrous, 99.8; Sigma-Aldrich), yttrium (III) nitrate tetrahydrate [Y(NO_3_)_3_∙II_2_O, 99.99% trace metal basis; Sigma-Aldrich] was dissolved, forming an Y_2_O_3_ precursor with 0.3 m concentration. The solution was sonicated for 20 min to obtain a transparent liquid. A portion of ITO was masked off with Kapton tape to serve as the contact area for the BE. The Y_2_O_3_ precursor was coated onto the prepared glass/ITO using a spin coater operating at 3000 rpm for a duration of 50 s. To vaporize the remaining solvent, the substrates were dried at 150 °C for 10 min on a hot plate in air (Corning stirrer hot plate, PC-420D, Corning, NY, USA). The annealing procedure was subsequently conducted to convert the Y_2_O_3_ precursor into Y_2_O_3_ films in a furnace at 500 °C for 2 h in air. A shadow metal mask with an electrode size of 30 µm × 30 µm was placed on the film, and a 100-nm-thick TE was deposited using a thermal evaporator at a vacuum level of 5 × 10^−n^ Torr and a deposition rate in the range of 1.8–2.0 Å/s. The structural features of the films were investigated using grazing incidence X-ray diffraction (GIXRD, X’pert Pro, Malvern PANalytical, Malvern, UK) with an incidence angle of 0.3° and a Cu Kα radiation wavelength of 1.54 Å. UV–visible spectrum analysis (LAMBDA 265, PerkinElmer, Waltham, MA, USA) was performed to evaluate the optical performance of the sintered oxide films at wavelengths ranging from 300 to 750 nm. Field-emission scanning electron microscopy (Hitachi 8230, Tokyo, Japan) and scanning probe microscopy (SPM; Park NX20, Park Systems, Suwon, Republic of Korea; tapping mode) were used to examine the surface roughness and thickness of the layer. The chemical composition of the films was analyzed using X-ray photoelectron spectroscopy (XPS; Nexsa, Thermo Fisher Scientific, Waltham, MA, USA) with a monochromatic Al Kα (1486.6 eV) source. The critical factors indicating the RRAM properties, namely the I–V curve, endurance, and retention were assessed using a probe station (MST T-4000A, Hwaseong, Republic of Korea) with a parameter analyzer (Keithley 2636B, Keithley, Cleveland, OH, USA). The represented performance parameters of RRAM devices were extracted from ten devices for five cycles per device.

## 3. Results and Discussion

Figure 1a displays the obtained GIXRD data from the annealed Y_2_O_3_ films. The diffraction peaks detected at 2*θ* values of 29.17°, 33.81°, 43.51°, 48.56°, and 57.66° are consistent with the Y_2_O_3_ crystallographic planes of (222), (400), (134), (440), and (622), respectively, confirming that the film has a polycrystalline structure. Through the most conspicuous peak at 29.17°, it can be deduced that grain growth dominantly occurred along the (222) plane direction. According to the GIXRD results, the observed peaks align well with the Y_2_O_3_ cubic structure (JCPDS: 89-5592), which is stable at relatively low temperatures [24]. Using the following Scherrer equation, the crystal size was computed:(1)$$D=\frac{0.9\lambda }{\beta \;cos\;\theta }$$
where *D*, *λ*, *β*, and *θ* denote the average crystal size, X-ray wavelength (1.54 Å), the full width at half maximum, and Bragg angle, respectively. The calculated crystal size of the Y_2_O_3_ films at the (222) plane was 10.97 nm. The surface roughness of the deposited Y_2_O_3_ films was obtained using SPM. The three-dimensional SPM surface image of the annealed film sized at 1 µm × 1 µm is presented in Figure 1b. The measured root-mean-square surface roughness (R_q_) for the films was 1.322 nm.

The chemical composition of the films was examined via XPS analysis. The acquired data were calibrated using the C1s peak at 284.5 eV. Figure 2a shows the Y3d peak, which consists of a doublet split into Y3d_5/2_ and Y3d_3/2_ states, located at 156.2 and 158.2 eV, respectively. This clearly indicates the formation of a Y–O bond. However, Y–OH bonds in the bulk of the film were also detected [25]. The O1s core-level spectra are presented in Figure 2b. The XPS peaks corresponding to pure Y (binding energies: 155.98 and 157.99 eV) did not appear [26]. The prepared precursors were successfully converted into Y_2_O_3_, not including any Y. These results also match well with the GIXRD data. The O1s spectra deconvoluted into five features at 528.9, 531.1, 532.1, 533.3, and 534.4 eV. The peaks are associated with M–O bonding in the Y_2_O_3_ lattice (O_L_), oxygen vacancy (O_V_), hydroxyl groups representing oxygen that is bound to the surface (–OH), O–C=O, and COOH, respectively [27,28]. When evaluating the relative area percentages of each component to the total area, the proportions of O_L_ and O_V_ are elucidated as 34% and 31%, respectively.

Figure 3a shows the transmittance of the glass, glass/Y_2_O_3_, and glass//ITO/Y_2_O_3_ films. The optical transparency of glass/Y_2_O_3_ films reaches 87% in the visible light region, extending from 400 to 700 nm in wavelength. The transmittance spectrum of the cleaned glass sample and glass/Y_2_O_3_ films are almost identical in the visible range, implying that sol–gel-processed Y_2_O_3_ films have a sufficient transparency. The remarkable transparency is attributed to almost negligible absorption losses, indicating the effective applicability of Y_2_O_3_ in electroluminescent applications. In contrast, the spectra of the glass/ITO/Y_2_O_3_ films exceed 76% in the visible light region. The decreased transmittance in the visible light region from 400 to 700 nm originates from the ITO BEs [29]. As the film thickness increases, the absorption increases. In these reasons, the transmittance spectrum of the cleaned glass sample and glass/Y_2_O_3_ films are almost identical in the visible range, due to the thin Y_2_O_3_ films. In contrast, in the case of glass/ITO/Y_2_O_3_ films, thick ITO films lead to a decreased transmittance [30,31]. The optical band gap was determined by graphically representing (*αhν*)*^n^* as a function of *hν* and performing linear fitting on the plotted graph. As shown in Figure 3b, the plot is derived using Tauc’s relation as follows:(2)$${\left(ahv\right)}^{n}=A(hv-E_{g})$$
where *α* is the absorption coefficient; *h* represents Planck’s constant; *ν* signifies the radiation frequency; and *n*, *A*, and *E_g_* stand for the parameter determined by the type of transition, a constant, and the optical band gap energy, respectively. When *n* equals 2, the case represents an allowed direct transition. Conversely, when *n* equals 1/2, the case represents an allowed indirect transition. The band gap for a direct transition was obtained from the x-intercept of the linear graph and was 4.11 eV, which is lower than the reported range of 5.5–6.0 eV for bulk Y_2_O_3_, as documented in the literature. The defect sites at the surface can broaden the valence band, as compared to bulk Y_2_O_3_. This results in a reduction in the optical band gap of the Y_2_O_3_ thin films [32].

Figure 4a illustrates the schematic structure of the fabricated Y_2_O_3_-based RRAM devices. Figure 4b–d show the representative I–V curves of the Y_2_O_3_-based stacked-structure RRAM devices with different TEs. To observe the RRAM characteristics, the BE was electrically grounded, and an external operating voltage was applied to the TE. DC voltage sweeps were conducted in ranges from −18.0 to +7.0 V and from +7.0 to −18.0 V. The electrical characteristics of the constructed Y_2_O_3_ RRAM devices showed different I–V curves with different TEs. First, the fabricated devices with the Au TE did not show any memory properties, but rather showed conventional insulator properties. Conversely, the Y_2_O_3_ RRAM devices with the Ag or Cu TE exhibited conventional bipolar memory properties, and the initial forming process was not needed [33,34]. The RRAM is classified into oxygen vacancy-based OxRRAM or metal ion-based CBRAM, according to the driving mechanism. As shown in Figure 4b, when using inert Au as the TE, resistive switching memory behavior was not observed; only insulating properties were observed. This implies that oxygen vacancies inside Y_2_O_3_ cannot trigger resistive switching memory characteristics. However, the RRAM characteristics were observed using Ag or Cu, which are electrochemically active metals. This indicates that the main mechanism of the fabricated Y_2_O_3_ RRAM devices is the growth of conductive filaments via the transport of metal ions and the redox processes, originating from the electrochemically active TEs. Additionally, these devices did not require an initial high voltage for the electroforming process, as shown in Figure 4c,d. The CBRAM with the Ag or Cu TE or the OxRRAM, consisting of oxygen vacancy-rich active channel layers, do not require the initial forming process because relatively conductive metal filaments or oxygen vacancy-based conductive filaments can be prepared comparatively much easier than other cases. The RRAM device initiates in the HRS. When a positive voltage is applied to the TE for the SET process, metal cations of the active electrode are generated at the Y_2_O_3_/TE interface because of the oxidation process (M → M^z+^ + z^e−^). The formed Ag^+^ or Cu^2+^ ions affected by an electric field drift toward the ITO BE, where the reduction of metal ions to Ag or Cu atoms occurs as they receive electrons from the BE (M^z+^ + z^e−^ → M). Finally, the metal atoms assemble and form conductive filaments within the Y_2_O_3_ film, connecting the TE and BE. On the basis of this process, the device demonstrates a switching behavior with an abrupt flow of current through this conductive path, transitioning from the initial HRS to the LRS. The positive voltage applied to form the conductive filament is referred to as the SET voltage. When a negative voltage is applied to the TE for the RESET process, the previously formed metal ions migrate back to the TE, causing the conductive paths to rupture. Consequently, the current decreases drastically, and the device returns to the HRS. The negative voltage applied to remove the conductive filament is referred to as the RESET voltage.

Figure 5 shows the extracted representative performance parameters of the RRAM devices, such as the voltages for SET and RESET and the HRS and LRS values. Except for the ITO/Y_2_O_3_/Au devices, the output characteristics of the Y_2_O_3_ RRAM devices are different, depending on the used TEs. First, the Y_2_O_3_ RRAM devices with the Ag TE showed a lower SET voltage, higher RESET voltages, and lower LRS values compared to the RRAM devices with the Cu TE. The formation of conductive filaments involves the following steps: (1) an oxidation process, (2) metal ion migration using a high electric field, and (3) a reduction process at inert BEs. On the basis of the model of Mott and Gurney, the ionic current density is provided using the following equation [21]: (3)$$i=2zecavexp\left(-\frac{W_{a}^{0}}{kT}\right)sinh(\frac{azeE}{2kT})$$
where *c* is the concentration of mobile cations M^z+^, *a* is the jumping distance of the ions, *v* is the voltage ramp rate, and *v* is the frequency factor. *W_a_* is the energy barrier for ion hopping. Furthermore, *k* is the Boltzmann’s constant, *T* is the absolute temperature, and *E* is the electric field (*E =*
$\Delta \emptyset_{SE}/d$). If *E* is much larger than *kT/aze* (*E >> kT/aze*), Equation (3) can be provided by
(4)$$i=zecavexp\left(-\frac{W_{a}^{0}}{kT}\right)exp(\frac{azeE}{2kT})$$

If *E* is much smaller than *kT/aze* (*E << kT/aze*), Equation (3) can be provided by
(5)$$i=\frac{{\left(ze\right)}^{2}cE}{kT}a^{2}vexp\left(-\frac{W_{a}^{0}}{kT}\right)=\frac{{\left(ze\right)}^{2}cD\Delta \varphi_{SE}}{kTd}$$
(6)$$D=a^{2}vexp\left(-\frac{W_{a}^{0}}{kT}\right)$$
where *D* is the diffusion coefficient. The high concentration of mobile cations M^z+^ caused by the oxidation process and the faster migration inside the main channel layer resulted in a faster transistor from the HRS to the LRS, as well as the formation of thicker conductive filaments. The standard redox potential (SRP) for Ag is larger than that for Cu. In an aqueous solution, the SRP values for Ag are 1.98 V (Ag^2+^/Ag^+^) and 0.8 V (Ag^+^/Ag), and those for Cu are 0.52 V (Cu^+^/Cu), 0.34 V (Cu^2+^/Cu), and 0.16 V (Cu^2+^/Cu^+^) [35]. This indicates that Cu can be easily oxidized compared to Ag, generating more mobile cations M^z+^. However, the Y_2_O_3_ RRAM devices with the Cu TE show a larger SET voltage, even though they have a larger mobile cation concentration. Even though the Ag mobile ion generation rate is lower, faster migration inside Y_2_O_3_ leads to a decrease in the SET voltage and the formation of thick conductive filaments. The thicker conductive filament also results in the formed conductive filament being more difficult to break, resulting in larger negative RESET voltages [36,37,38].

A greater difference in the work function between the TE and the BE is a factor that enables easier filament formation. This is attributed to the built-in field created by the difference in the work function, which facilitates the movement of M^z+^ ions toward the cathode [39,40]. The work functions of Ag, Cu, and ITO are 4.26, 4.65, and 4.7 eV, respectively. The work function difference between Ag and ITO is −0.44 eV (4.26–4.7 eV), and that between Cu and ITO is −0.05 eV (4.65–4.7 eV) [36,41]. The built-in field of the device was determined using the following expression [40]:(7)$$E_{bi}=\frac{\Delta \Phi }{q\times t_{ox}}$$
where *E_bi_* represents the built-in field, Δ*Φ* is the difference between the work function of the TE and BE, *q* is the charge amount, and *t_ox_* is the oxide thickness (55 nm). The *E_bi_* obtained for the Ag device was 0.8 × 10^5^ V·cm^−1^, while that for the Cu device was 0.09 × 10^5^ V·cm^−1^. The effective field at the SET voltage was obtained using the following equation:(8)$$E_{eff}=E_{bi}+\frac{V_{set,a}}{t_{ox}}$$
where *E_eff_* and *V_set_,_a_* correspond to the effective field at the SET voltage and the average value of the SET voltage (*V_set,a_* of Ag = 2.7 V and *V_set,a_* of Cu = 5.2 V), respectively. For the Ag device, *E_eff_* was approximately 0.57 × 10^6^ V·cm^−1^, and for the Cu device, it was approximately 0.95 × 10^6^ V·cm^−1^ [40]. Through this, it can be observed that the *E_eff_* of Ag is lower than that of Cu, causing ITO/Y_2_O_3_/Ag to be fixed at a lower voltage than ITO/Y_2_O_3_/Cu. The energy diagrams of Ag/Cu, Y_2_O_3_, and ITO before and after contact are shown in Figure 6.

As shown in Figure 7a–d, the properties of the nonvolatile RRAM devices were assessed as a function of the pulse width (0.05, 0.5, and 5 s) via measuring the endurance and retention performance at a read voltage of 0.1 V. For programming and erasing, +7.0 and −18.0 V were applied, respectively. As the pulse width increases, the fabricated Y_2_O_3_ RRAM devices have sufficient time for the oxidation process at the TEs and Y_2_O_3_, as well as to migrate to the inert BEs for each cycle. The obtained average number of cycles increases from 99 to 133 for the ITO/Y_2_O_3_/Ag RRAM devices and from 7 to 15 for the ITO/Y_2_O_3_/Cu RRAM devices. However, at this average number of cycles under the same biased pulse width condition, the ITO/Y_2_O_3_/Cu RRAM devices show worse endurance properties, which originate from the slow migration speed. This makes it difficult to form the conductive filament during the programming process. A larger voltage or a longer pulse width may be required to form the conductive filaments for the ITO/Y_2_O_3_/Cu RRAM devices. In contrast, the retention data in the two cases indicate that the resistance states remained stable up to 10^4^ s without significant degradation. Compared to the modern devices, still operation voltages are high, and endurance and retention properties are relatively poor. O_V_ concentration control in metal oxides or local electric field generation help to reduce the operation voltage and enhance endurance and retention properties by suppressing randomly formed conductive filaments. These strategies are needed to improve device performance [42,43].

## 4. Conclusions

Y_2_O_3_ RRAM devices were fabricated using the sol–gel process. For the TEs, electrochemically inert Au and electrochemically active Ag and Cu were used. The Y_2_O_3_ RRAM devices fabricated by the deposition of the inert Au TE did not exhibit conventional RRAM characteristics. In contrast, the Y_2_O_3_ RRAM devices with Ag or Cu, which are electrochemically active TEs, showed conventional bipolar memory characteristics. Conductive metal filaments were more easily formed using Ag TEs when compared to the Cu TE case. The fast migration of metal ions inside the Y_2_O_3_ active channel layer and improved built-in field led to a lower SET voltage, and also encouraged the growth of thick conductive filaments. The easy formation of conductive metal filaments also improves the number of programming–erasing cycles. Relatively, the formation of conductive metal filaments is harder when using Cu TEs, thus resulting in a larger SET voltage and a degradation of the programming–erasing cycle characteristics (endurance). The electrochemical properties of the TEs for CBRAM play a critical role in determining the electrical characteristics of RRAM devices. These results contribute to a deeper understanding of how these factors influence the resistive switching behavior of metal oxide-based atomic switches and CBRAM cells.

## Figures and Tables

**Figure 1 nanomaterials-14-00532-f001:**
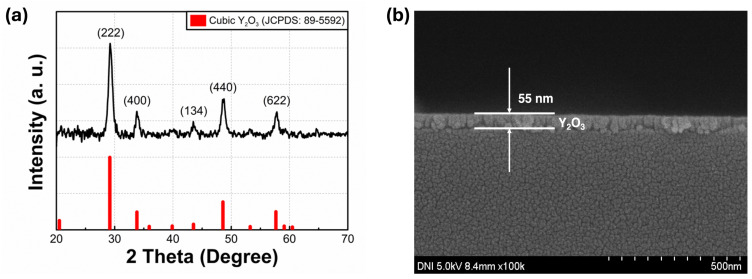
(**a**) GIXRD spectra and (**b**) the cross-sectional SEM image of Y_2_O_3_ film. Three-dimensional SPM surface images of the ITO/glass substrates, (**c**) and a sol–gel-processed Y_2_O_3_ film on ITO/glass substrates (**d**), respectively.

**Figure 2 nanomaterials-14-00532-f002:**
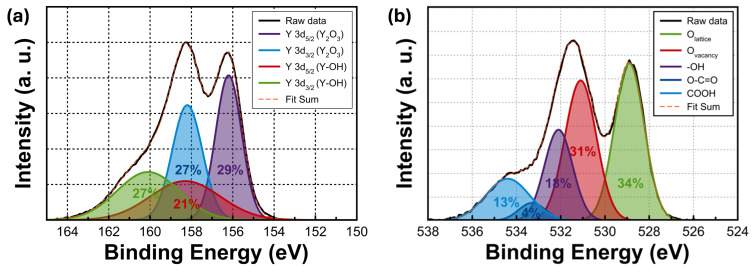
XPS curves of (**a**) Y3d and (**b**) O1s for the sol–gel-processed Y_2_O_3_ films.

**Figure 3 nanomaterials-14-00532-f003:**
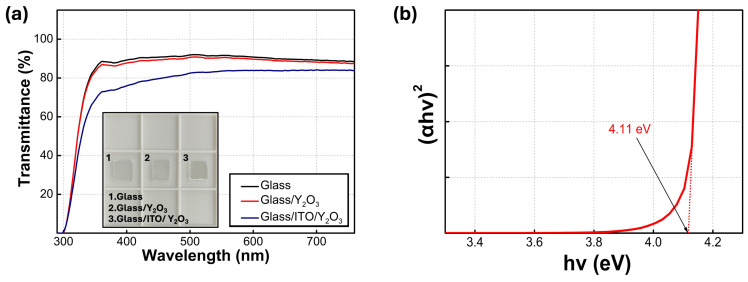
(**a**) Transmittance spectra of bare glass, glass/Y_2_O_3_ films and the glass/ITO/Y_2_O_3_; the inset shows the optical images of the films. (**b**) (*αhν*)^2^-versus-*hν* plot of the Y_2_O_3_ films.

**Figure 4 nanomaterials-14-00532-f004:**
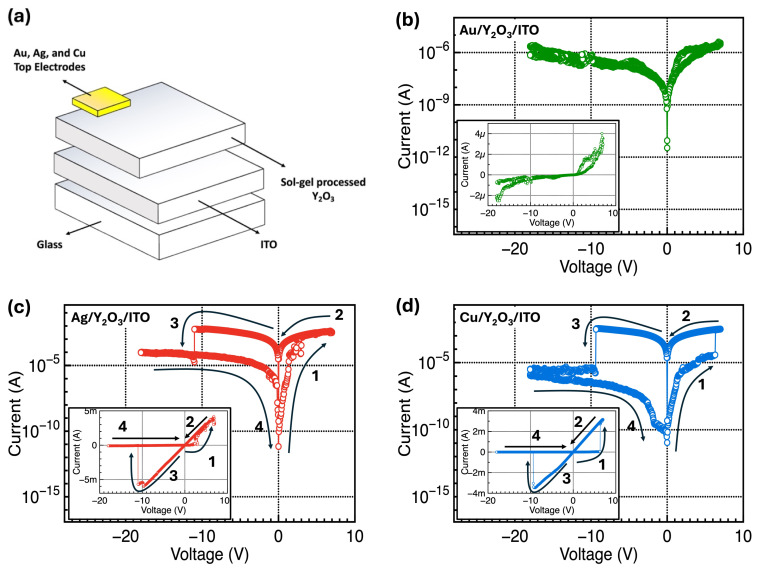
(**a**) Schematic of the fabricated Y_2_O_3_-based RRAM devices and (**b**–**d**) representative I–V curves of the Y_2_O_3_-based stacked-structure RRAM devices with different TEs.

**Figure 5 nanomaterials-14-00532-f005:**
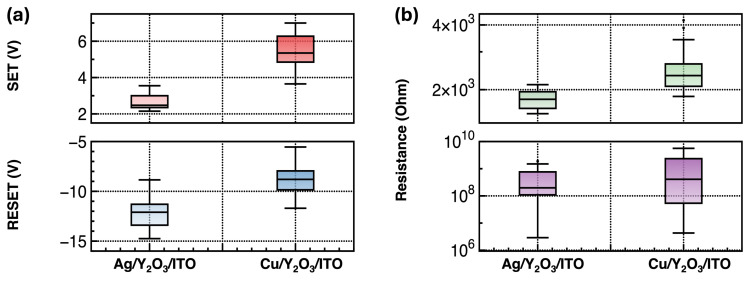
Extracted performance parameters of the RRAM devices: (**a**) SET and RESET voltages and (**b**) LRS and HRS values for ten devices, with five cycles per device.

**Figure 6 nanomaterials-14-00532-f006:**
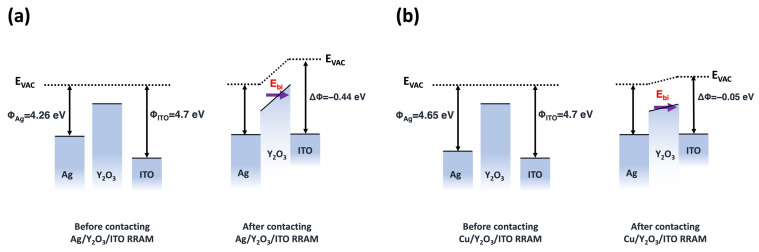
Energy diagrams for (**a**) Ag/Y_2_O_3_/ITO and (**b**) Cu/Y_2_O_3_/ITO RRAM devices, before and after contact.

**Figure 7 nanomaterials-14-00532-f007:**
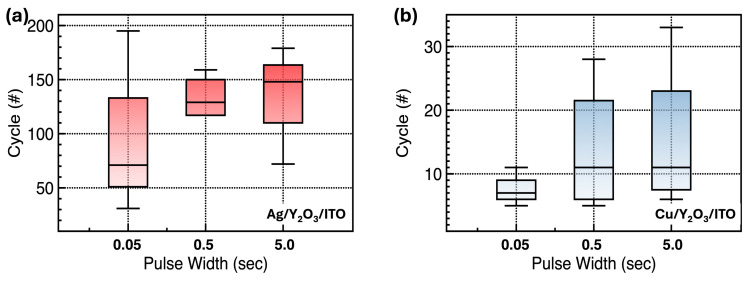
Endurance test results as a function of pulse width: (**a**) ITO/Y_2_O_3_/Ag and (**b**) ITO/Y_2_O_3_/Cu RRAM devices as a function of pulse widths. Retention test results: (**c**) ITO/Y_2_O_3_/Ag and (**d**) ITO/Y_2_O_3_/Cu RRAM devices, after the programming and erase operations (each pulse width: 50 ms).

## Data Availability

Data are contained within the article.

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
