# Peer review of "Effect of Electrochemically Active Top Electrode Materials on Nanoionic Conductive Bridge Y2O3 Random-Access Memory"

_nanomaterials, 2024, doi:10.3390/nano14060532_

Round 1

Reviewer 1 Report

Comments and Suggestions for Authors

1.Introduction:The introduction of this article does not have any review for Y2O3 for application in RRAM devices , Pleasr joint it , and please explain why the s0l-gel mrthod is used.

2,Experimental:Experimental materials(sources) and methods need to be explained in more  details 

ex.SPM type , material sources e.t.c.

3.Results and discussion:

Fig1(a) XRD base line must be correctted.

Fig1(b) SPM please 3-D diagram

Transparaency

more details see the attachment.

Comments on the Quality of English Language

mino modification

Reviewer 2 Report

Comments and Suggestions for Authors

The article is devoted to the possibility of creating memory devices based on a layer of yttrium oxide applied by the sol-gel method. First of all, I would note the low information content of the literature review. Instead of considering the specific characteristics of known devices, the authors simply list device types and materials. References are given to many sources at once [6-16] without much analysis. I note that about 25% of the cited articles belong to the authors of this work (5-10, 14-16, 20). Overall, I consider the literature review to be unsatisfactory. The idea of forming high-speed memory elements with low switching voltages using the sol-gel method also seems dubious. In general, it seems to me that this work is still far from being relevant to modern trends in this direction. Devices produced by the sol-gel method apparently cannot compete with other methods. Special requirements are imposed on nanoscale devices. Apparently the obtained levels of roughness and thickness of the Y2O3 layer will not allow scaling this technology to nanoscale sizes.

Despite my negative opinion, I will still make the remark that the research part as a whole is written very well. The data on X-ray measurements and on the composition of the films are very interesting and relevant, carried out using adequate methods.

The methodology is adequate, except that switching speed has not been studied.

There are few comments 

1. Line 119. "The measured root-mean-square roughness forthe films was 1.322 nm. " It would be useful to provide data on the roughness of the original substrate and the ITO layer.

2. I didn't fing Y2O3 thickness  in the paper. And how it was measured. Probably it stated at line 260 (55nm), but seems unclear.

3. Actually set and reset voltages are to high to application in intergated memory devices. The authors should state how the voltage can be decreased.

4. Tha main drawback of the devices if low endurance and retention performance. Actually it seem far beyond any comparison with  any kind of modern devices for RAM. Also time of set/reset switching wasnt measured.

Reviewer 3 Report

Comments and Suggestions for Authors

This manuscript is to introduce sol-gel spin-coated Y2O3 used for resistive random-access memory (RRAM) devices. Similar to their previous studies, the device was prepared well along with required characterizations. Here is a summary of minor corrections to improve the quality of the manuscript.

1. Equations should be carefully confirmed, including notations, after defining all symbols. Also, figures should be double-checked.  In Figure 4, the units of insects for IV curves should be changed, not m or m.

2. Some abbreviations, such as SET and RESET, were poorly defined.

Round 2

Reviewer 1 Report

Comments and Suggestions for Authors

Authors have been revised all requires

Reviewer 2 Report

Comments and Suggestions for Authors

The authors kindly adressed all of my notes and remarks. Now i brlieve that the paper present research or proper quality and fine soundness. I think that the material under study is seemingly far from topical microelectronic quality and far from the best technological examples for RRAM.  The paper is written very well. The data on X-ray measurements and on the composition of the films are very interesting and relevant, carried out using adequate methods. The methodology is adequate. I think paper can be accepted in the present form.